# H1N1 Influenza A Virus Protein NS2 Inhibits Innate Immune Response by Targeting IRF7

**DOI:** 10.3390/v14112411

**Published:** 2022-10-31

**Authors:** Bo Zhang, Minxuan Liu, Jiaxin Huang, Qiaoying Zeng, Qiyun Zhu, Shuai Xu, Hualan Chen

**Affiliations:** 1College of Veterinary Medicine, Gansu Agricultural University, Lanzhou 730070, China; 2State Key Laboratory of Veterinary Etiological Biology, Lanzhou Veterinary Research Institute, Chinese Academy of Agricultural Sciences, Lanzhou 730046, China

**Keywords:** influenza A virus, nonstructural protein 2, interferon, interferon regulatory factor 7

## Abstract

Influenza A virus (IAV) is a globally distributed zoonotic pathogen and causes a highly infectious respiratory disease with high morbidity and mortality in humans and animals. IAV has evolved various strategies to counteract the innate immune response, using different viral proteins. However, the mechanisms are not fully elucidated. In this study, we demonstrated that the nonstructural protein 2 (NS2) of H1N1 IAV negatively regulate the induction of type-I interferon. Co-immunoprecipitation experiments revealed that NS2 specifically interacts with interferon regulatory factor 7 (IRF7). NS2 blocks the nuclear translocation of IRF7 by inhibiting the formation of IRF7 dimers, thereby prevents the activation of IRF7 and inhibits the production of interferon-beta. Taken together, these findings revealed a novel mechanism by which the NS2 of H1N1 IAV inhibits IRF7-mediated type-I interferon production.

## 1. Introduction

Influenza A virus (IAV) is a serious public health concern that causes annual seasonal epidemics and even sporadic pandemics, which is a continual threat to animal and human health. Since the Spanish flu pandemic of 1918, IAV has caused four influenza pandemics [1,2,3]. Additionally, the rapid and frequent mutation and recombination of influenza virus caused the emergence of highly pathogenic H5N1, H5N6, H5N8, and H7N9 IAVs, leading to a significant loss of human life [4,5,6,7]. IAVs are members of the *Orthomyxoviridae* family that are segmented, single-stranded, and negative-sense RNA viruses. The IAV genome consists of eight gene segments, comprising basic polymerase 2 (PB2), basic polymerase 1 (PB1), acidic polymerase (PA), hemagglutinin (HA), nucleoprotein (NP), neuraminidase (NA), matrix (M), and nonstructural protein (NS), which encode at least 14 proteins [8,9].

The NS gene encodes nonstructural protein 1 (NS1) and nonstructural protein 2 (NS2) by mRNA splicing. NS1 inhibits both the production of interferon and the induction of several interferon-induced genes [10]. Moreover, NS1 plays a role in the packaging of virions, as well as inhibiting the processing and nuclear export of host mRNAs [11]. In addition to NS1, viral segment NS encodes a 121-amino-acid polypeptide, which was originally thought to have no structural function within the virion, leading to its designation as NS2 [12,13]. Nowadays, NS2 has been shown to have several important functions, such as modulating the accumulation of viral RNA during transcription and translation [14,15,16] and interacting with host proteins such as ATPase and nucleoporins to benefit viral budding [17,18]. Moreover, NS2 plays a role in regulating the innate immune system.

In the early stages of infection, the innate immune system uses pattern recognition receptors (PRRs) to recognize the conserved pathogen-associated molecular patterns (PAMPs) of invading pathogens [19,20]. Upon IAV infection, retinoic acid-inducible gene-I (RIG-I) recognizes viral RNA, interacts with the mitochondrially localized adaptor protein MAVS, and finally activates interferon regulating factor 3 (IRF3) and interferon regulator factor 7 (IRF7) that were phosphorylated and translocated into nucleus to activate the transcription of type-I interferons (IFN-I), and ultimately create an antiviral state [21,22]. In this process, as primary transcription factors, IRF3 and IRF7 bind to the promotor regions of interferon gene to regulate IFN-I production [23,24]. To effectively infect and replicate in host cells, viral proteins of IAV play various roles to counteract and evade host defenses [25,26,27,28]. Among them, IAV NS1 is a well-studied antagonist of the induction of IFNs that facilitate viral replication or enhance pathogenicity in animals via distinct mechanisms [10,29,30]. Recently, Liu et al. have reported that H1N1 IAV NS2 interacts with a member of the histone H1 family, HIST1H1C (H1C), to reduce H1C-IRF3 interaction, and results in the inhibition of IFN-I [31]. However, the molecular mechanism of NS2-mediated regulation of the IFN-I signaling pathway is not yet fully understood.

In this study, we demonstrated that the NS2 protein of H1N1 IAV negatively regulate the induction of IFN-I. NS2 prevents the activation of IRF7 and inhibits the production of interferon-beta (IFN-β). Specifically, NS2 inhibits the formation of IRF7 dimers, thereby blocking the nuclear translocation of IRF7. Our findings reveal a novel mechanism used by H1N1 IAV for the evasion from host antiviral immune responses.

## 2. Materials and Methods

### 2.1. Biosafety and Ethical Statements

This study was performed strictly following the recommendations in the Guide for the Care and Use of Laboratory Animals of the Ministry of Science and Technology of the People’s Republic of China. Studies with influenza viruses were conducted in a biosafety level 2 laboratory approved for such use by the Lanzhou Veterinary Research Institute, Chinese Academy of Agricultural Sciences.

### 2.2. Cells, Viruses, and Plasmids

HEK293 and A549 cells were obtained from ATCC, and U2OS cells were kindly provided by Dr. Cao-Qi Lei (Wuhan University). HEK293 and U2OS cells were grown in DMEM (Gibco, Waltham, MA, USA) supplemented with 10% (*v*/*v*) FBS (Gibco, Waltham, MA, USA) and 1× penicillin/streptomycin (Gibco, Waltham, MA, USA). A549 cells were grown in Kaighn’s modified Ham F-12 nutrient mixture medium (Gibco, Waltham, MA, USA) supplemented with 10% FBS and penicillin/streptomycin. All cells were cultured and maintained at 37 °C with 5% CO_2_.

Influenza A virus, A/Puerto Rico/8/1934 (PR8, H1N1) (Gene ID: 3926091), and Sendai virus (SeV) were inoculated into 10-day-old specific-pathogen-free (SPF) chicken embryos for virus propagation and stored at −70 °C as virus stocks until use.

Plasmids for HA-, Flag-tagged RIG-I-CARD (RIG-IN), MDA5-CARD (MDA5-N), MAVS, TBK1, IKKε, IRF3, IRF7, IRF9 and the IFN-β-Luc, ISRE-Luc, STAT1-Luc and pRL-TK internal control luciferase reporter plasmids used in the study were described previously [32,33]. The NS2 genes from H1N1, H5N1, H7N9, and H9N2 viruses were synthesis by Tsingke (Beijing, China) and inserted into the pRK expression vector with Flag-tag at the N-terminus. GFP-tagged NS2, NS2-N, and NS2-C were constructed and cloned into pEGFP-C1 by using standard molecular biology techniques.

### 2.3. Reagents and Antibodies

The antibodies used in this study were as follows: HRP-conjugated anti-HA (12013819001) and HRP-conjugated anti-Flag (A8592) antibodies (Sigma, Dallas, TX, USA); anti-GFP (11814460001), TBK1 (3013S) antibodies (Cell Signaling Technology, Danvers, MA, USA); anti-NS2 (GTX125953) antibody (Genetex, Irvine, CA, USA); anti-IRF7 (ab238137), anti-GAPDH (ab181602), and HRP-conjugated mouse anti-rabbit IgG (light-chain specific) (93702) antibodies (Abcam, Boston, USA); Alexa Fluor 488-conjugated anti-mouse IgG (A0428), Cy3-labeled goat anti-rabbit IgG (A0516) (Beyotime, Shanghai, China).

Reagents used in the study included: anti-Flag agarose affinity beads (A2220) and protein A/G agarose affinity beads (P6486/E3403) (Sigma, Dallas, TX, USA); human IFN-β DuoSet ELISA kit (DY814-05) (R&D, Minneapolis, MN, USA); NP-40 (ST366), DAPI (C1002), and Passive Lysis 5× Buffer (E1941) (Promega, Madison, WI, USA). The scrambled negative control RNA (NC), and IRF7-specific short interfering RNA were purchased from RiboBio Co. (Guangzhou, China). Lipofectamine 2000, RNAi MAX, and TRIzol were obtained from Invitrogen (Carlsbad, CA, USA). SYBR Green I Master Mix was purchased from Roche (Darmstadt, Germany).

### 2.4. Dual-Luciferase Reporter Assays

To detect activation of the IFN-I pathway, HEK293 cells grown in 24-well plates were co-transfected with luciferase reporter plasmids (IFN-β-Luc, ISRE-Luc or STAT1-Luc) and the pRL-TK plasmid, along with the indicated amounts of empty vector (Vec) or plasmids expressing NS2 or other molecules [33]. At 24 h post-transfection, cells were stimulated by SeV for 12 h or IFN-β for 4 h. Cell lysates were then prepared and analyzed for Firefly and *Renilla* luciferase activities using the dual luciferase assay kit (Promega, Madison, USA).

### 2.5. RNA Isolation and Quantitative PCR

Total RNA from cells was extracted with TRIzol following the manufacturer’s instructions. For mRNAs, total RNA was subsequently transcribed into cDNA using M-MLV Reverse Transcriptase, according to the manufacturer’s protocol (Promega, Madison, WI, USA). GAPDH was used as an invariant control for mRNAs. Real-time PCR was carried out using the ABI 7500 Detection System (Applied Biosystems, Carlsbad, CA, USA). The RNA level of each gene is shown as the fold of induction (2^−ΔΔCT^) in the graphs. The sequences of gene-specific primers used for qPCR are provided in Appendix A.

### 2.6. Western Blot

Cells were lysed in RIPA buffer (Beyotime, Shanghai, China). Proteins were then separated on 10% SDS-PAGE gels and transferred to nitrocellulose membrane (Bio-Rad, Hercules, CA, USA). Membranes were blocked for 1 h in TBST containing 5% skim milk and subsequently incubated with primary antibodies overnight at 4 °C. After incubation with HRP-conjugated secondary antibodies for 1 h, immunoreactive bands were visualized using an ECL system (GE Healthcare, Fairfield, CT, USA).

### 2.7. Co-Immunoprecipitation

HEK293 cells or A549 cells were first co-transfected with indicated plasmids with or without virus infection for 24 h. The transfected cells were then harvested and lysed in NP-40 lysis buffer (20 mM Tris-HCl (pH 7.5), 150 mM NaCl, 1% NP-40, 1 mM EDTA, and protease inhibitor cocktail). For each immunoprecipitation, 1 mL of lysate was incubated for 4 h at 4 °C with 0.5 μg of indicated antibody or control IgG and 30 μL of protein A/G-Sepharose (Sigma, Dallas, TX, USA). The beads were then washed three times with 1 mL of lysis buffer containing 500 mM NaCl. The precipitates were finally analyzed using standard immunoblotting.

### 2.8. Confocal Microscopy

Confocal microscopy was performed as previously described [34]. Cells were seeded in 12-well plates on coverslips, and transfected for 24 h before harvest. Cells were then fixed with 4% paraformaldehyde for 20 min at room temperature, and washed three times with PBS. Cells were permeabilized with 0.1% Triton X-100 in PBS for 10 min and blocked with 5% skim milk for 1 h. Then, the cells were incubated with indicated primary and secondary antibodies and DAPI. The stained cells were observed with a Leica microscope (TCS SP8) with a 100× oil objective (NA 1.40).

### 2.9. VSV-GFP Bioassay

Antiviral cytokine secretion bioassays were conducted as with slight modifications [32]. In brief, HEK293 cells were grown in 24-well plates and transfected with indicated plasmids. At 24 h post-transfection, cells were infected with SeV (MOI = 1) or not (Mock) for another 24 h. Cell supernatants were harvested and inactivated by placing samples on ice and illuminated with ultraviolet radiation for 20 min. The ultraviolet radiation-inactivated supernatants were added to fresh confluent HEK293 cells, and these cells were then incubated for another 24 h. The cells were then infected with VSV-GFP (MOI = 0.01). At 12 h post-infection, VSV-GFP replication was visualized by monitoring the GFP expression level by fluorescence microscopy.

### 2.10. Statistical Analysis

Data are expressed as mean ± standard deviation (SD). Statistical significance was determined by using Student’s two-tailed non-parametric *t* test or analysis of variance (ANOVA) with GraphPad Prism software (version 6.0, San Diego, CA, USA). Differences between groups were considered significant when the *p* value was <0.05 (*), <0.01 (**), and <0.001 (***).

## 3. Results

### 3.1. NS2 Inhibits IFN-I Induction

It is well known that during the viral replication cycle, NS2 acts as an adaptor to mediate the export of viral ribonucleoproteins (vRNPs) from the nucleus to the cytoplasm [35,36,37]. Nevertheless, little is known about the role of NS2 in the process of viral escape from host antiviral innate immunity. In this study, we sought to explore the role of NS2 in virus-triggered IFN-β activation. As shown in Figure 1A, overexpressed H1N1-NS2 strongly inhibited SeV-triggered activation of the IFN-β promoter and ISRE, but NS2 had no inhibitory effects on the IFN-β-triggered activation of STAT1-luciferase reporter, which contains STAT1 response element. Consistently, overexpressed NS2 from different IAV subtypes strongly inhibited SeV-triggered activation of the IFN-β promoter (Figure 1B). Furthermore, H1N1-NS2 significantly inhibited the production of IFN-β triggered by overexpressed RIG-IN and SeV infection (Figure 1C). Quantitative PCR (qPCR) analysis indicated that overexpression of NS2 significantly inhibited SeV-triggered transcription of the interferon beta 1 (*IFNB1*), IFN-stimulated gene 15 (*ISG15*), IFN-stimulated gene 56 (*ISG56*), interferon-induced protein with tetratricopeptide repeats 2 (*IFIT2*), and regulated upon activation normal T cell expressed and secreted factor (*RANTES*) genes in A549 cells (Figure 1D). Since NS2 inhibited virus-triggered induction of downstream effector genes, we next investigated whether NS2 plays a role in the cellular antiviral response. As shown in Figure 1E, supernatants from SeV-infected and MAVS-overexpressed cells completely inhibit the replication of GFP-expressing vesicular stomatitis virus (VSV-GFP) in fresh cells as monitored by GFP fluorescent intensity, while NS2- and NS1-transfected cells stimulated with SeV could not completely inhibit the replication of VSV-GFP, suggesting that NS2 dampened the secretion of antiviral factors induced by SeV. Taken together, these data suggested that the NS2 protein of IAV inhibits the induction of IFN-I.

### 3.2. NS2 Interacts with IRF3 and IRF7

Next, we investigated how NS2 inhibits IFN-I induction. To determine the target of NS2 in the regulation of virus-triggered IFN-β induction, we transfected plasmids encoding RIG-IN, MDA5-N, MAVS, TBK1, IKKε, IRF3, or IRF7 together with the IFN-β promoter in the presence or absence of NS2. We found that the overexpression of NS2 inhibits the activation of the IFN-β promoter triggered by the expression of RIG-IN, MDA5-N, MAVS, TBK1, IKKε, IRF3, and IRF7 in a dose-dependent manner, but it had no inhibitory effects on the activity of STAT1-luciferase triggered by IRF9 (Figure 2A). Co-immunoprecipitation (Co-IP) assays indicated that Flag-tagged NS2 interacted with HA-tagged IRF3 and IRF7, but not with other tested molecules (Figure 2B). Confocal microscopy confirmed that NS2 co-localized with IRF3 and IRF7 (Figure 2C). These data indicated that IRF3 and IRF7 might be the targets of NS2 to inhibit the IFN-I induction.

### 3.3. NS2 Targets IRF7 and Inhibits IFN-I Production

Liu et al. has reported that histone H1C regulates IFN-I production by interacting with IRF3, and NS2 reduces H1C-IRF3 interaction to inhibit the H1C-enhanced IFN-β expression [31]. Considering this, we started to investigate the role of IRF7 in the NS2-regulated IFN-I expression. Following H1N1 virus infection, endogenous Co-IP experiment verified that NS2 was associated with IRF7 in the infected cells, with TBK1 as a positive control to interact with IRF7 (Figure 3A). Luciferase and ELISA assays indicated that overexpression of NS2 significantly inhibits SeV-mediated activation of the IFN-β promoter and the induction of IFN-β, but in cells transfected with siRNA of IRF7, NS2 could not inhibit the activation of the IFN-β promoter and the production of IFN-β (Figure 3B,C). Collectively, the data revealed that IRF7 is another specific target of NS2 to inhibit the IFN-I induction.

### 3.4. NS2 Blocks the Nuclear Translocation of IRF7

Viral infection leads to IRF7 activation and translocation into the nucleus, where it binds to the promoter regions of interferon gene to activate the transcription [38]. We then examined the possibility that NS2 alters the ability of IRF7 to enter the nucleus. As shown in Figure 4A, the NS2 localized in both nucleus and cytoplasm, and the IRF7 localized largely in the cytoplasm of mock-treated cells. Following SeV stimulation, IRF7 translocated into nucleus, but the expression of NS2 prevented the nuclear translocation of IRF7. Using subcellular fraction isolation, we further analyzed the expression of IRF7 in cytoplasmic and nuclear fractions, and detected that NS2 reduces the expression of IRF7 in the nucleus of SeV-treated cells (Figure 4B). Using native-PAGE to analyze the dimerization of IRF7, we found that SeV treatment promotes the formation of IRF7 dimers, and overexpressed NS2 significantly inhibited the dimerization of IRF7. As a verification, we co-transfected an increasing amount of NS2 together with HA- and Flag-tagged IRF7 and performed Co-IP assays. We found that Flag-NS2 interacts with HA-IRF7, and dose-dependently inhibits the interaction between HA-IRF7 and Flag-IRF7, which indicated that NS2 bound to HA-IRF7 in competition with Flag-IRF7 in a dose-dependent manner. Together, these results suggested that NS2 influences the formation of IRF7 dimers by competing with IRF7, and further blocks the nuclear translocation of IRF7.

### 3.5. The N-Terminal Domain of NS2 Is Essential for the Interaction with IRF7

To explore which domain of NS2 was involved in the interaction with IRF7, we constructed two vectors expressing the two domains of NS2 (Figure 5A), the N-terminal domain (NS2-N, amino acids (aa) 1 to 53) and the C-terminal domain (NS2-C, aa 54 to 121) [17]. HEK293 cells were transfected with GFP-NS2, GFP-NS2-N, or GFP-NS2-C along with HA-IRF7, and Co-IP assays were performed with anti-HA and anti-GFP antibodies. We found that the N-terminal domain of NS2 was immunoprecipitated by IRF7, and reciprocally, IRF7 could be immunoprecipitated by the N-terminal domain of NS2, whereas the C-terminal domain of NS2 did not interact with IRF7 (Figure 5B,C). As shown in Figure 5D, the N-terminal domain of NS2 inhibited the production of IFN-β triggered by RIG-IN and SeV. Therefore, the N-terminal domain of NS2 was the critical domain for the interaction with IRF7 and inhibition of IFN-β production.

## 4. Discussion

In the present study, we found that inhibition of IFN-β production mediated by NS2 was common to several different subtypes of IAVs. The results indicated that NS2 inhibits IFN-I signaling and IFN-β production. Co-IP experiments indicated that HA-tagged IRF3 and IRF7, but not other tested molecules, and specifically interacts with Flag-tagged NS2. Endogenous Co-IP experiments further verified that NS2 associates with IRF7 following H1N1 virus infection. Mechanistically, NS2 blocks the nuclear translocation of IRF7 by inhibiting the formation of IRF7 dimers, thereby prevents IRF7 activation and inhibits IFN-β production. Our study revealed a novel mechanism by which NS2 negatively regulates the innate immune system.

During the IAV cycle, viral proteins need to perform various functions to infect and replicate in the host. Numerous studies have shown that NS2 plays multiple roles in the replication cycle of IAV. As one of the non-structural proteins of IAV, NS2 is also known as nuclear export protein (NEP). NS2 acts as an adaptor to mediate the nuclear export of vRNPs by forming the Crm1-NS2-M1-vRNP complex [39,40]. Additionally, NS2 promotes the efficient release of budding virions by recruiting F1Fo ATPase [18]. Moreover, adaptive mutations in NS2 can increase viral RNA accumulation, which compensates for the reduced activity of avian viral polymerase in mammalian cells, thereby allowing highly pathogenic avian H5N1 influenza viruses to overcome host restrictions [41]. However, our study found that NS2 has an additional function of regulating the innate immune response.

Innate immune is the first line of host defenses against viral invasion [42]. Viruses employ a variety of strategies to fight and evade the innate defenses. Strategies used by different viruses include the evasion of recognition, the cleavage or degradation of essential innate immune molecules, and the blockade of molecular interactions [28,43,44,45,46]. Various structural and nonstructural viral proteins in IAVs have been reported to function as innate immune modulators which inhibit host immunity, like PB2, PB1, NP, NS1, PA-X, and PB1-F2 [32,47,48,49,50,51]. Similarly, as a non-structural protein of IAV, NS2 has the function of regulating the innate immune response. A previous study reported that the NS2 of IAV interacts with H1C and reduces H1C-IRF3 interaction, leading to the inhibition of IFN-β [31]. In the present study, we confirmed that NS2 inhibits IFN-β production, and this effect was common to several different subtypes of IAVs (Figure 1B). Importantly, NS2 competitively interacted with IRF7 (Figure 4D).

IRF7, as a crucial transcription factor, plays an essential role in host antiviral innate immunity [52]. Following virus infection, TBK1 phosphorylates IRF7, resulting in IRF7 dimerization and migration to the nucleus, where it binds to the IFN-β promoter [53,54,55]. Therefore, viruses have developed various strategies to counteract the activation of IRF7. On the one hand, the studies of DNA viruses revealed that viral proteins interact with IRF7 and regulate its function. Lin et al. has reported that the HSV-1 ICP0 protein inhibits IRF7 phosphorylation by targeting TBK1 and IKKε [56]. One study has shown that the Epstein–Barr virus LF2 tegument protein specifically interacts with the central inhibitory association domain of IRF7, leading to the inhibition of IRF7 dimerization [57]. ORF45, as KSHV immediate early protein, blocks the phosphorylation and nuclear accumulation of IRF7 during viral infection [58]. On the other hand, in the research of RNA viruses, the interaction between viral proteins and IRF7 has also been demonstrated [59,60,61]. Our findings further extend the understanding that NS2 of IAV blocks the nuclear translocation of IRF7 (Figure 4A), through preventing IRF7-dimer formation (Figure 4C). As a result, NS2 inhibits the activation of IRF7, and negatively regulates the IFN-I induction. Lastly, we identified the N-terminal domain of NS2 as the key domain for the interaction with IRF7 (Figure 5B,C). Nevertheless, we did not map any specific domain(s) in IRF7 that may involve in the interaction with NS2, which could be further investigated.

## 5. Conclusions

In summary, we uncovered a negative regulatory mechanism involving the NS2-IRF7 axis to evade the innate immune response, which provides valuable insights into the evasion strategy against innate signaling employed by influenza A virus.

## Figures and Tables

**Figure 1 viruses-14-02411-f001:**
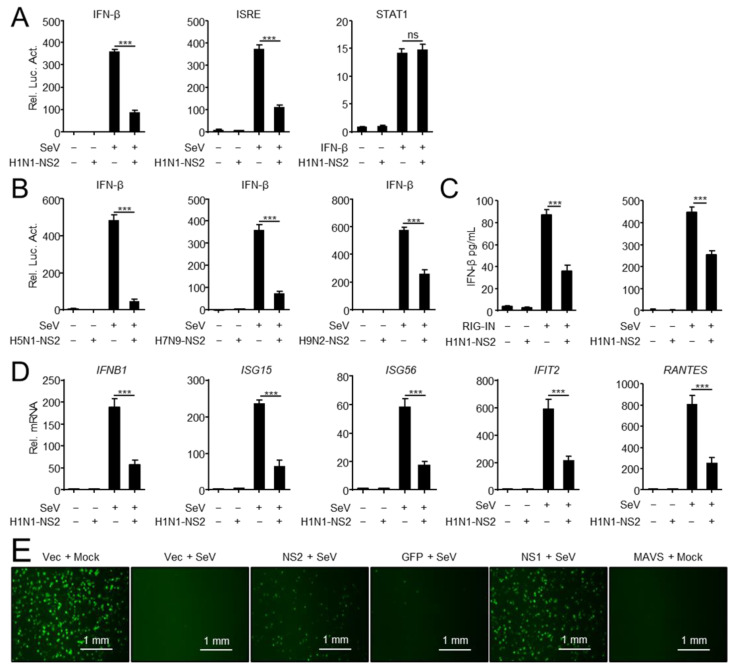
NS2 inhibits IFN-I induction. (**A**) NS2 specifically inhibits SeV-triggered activation of IFN-β and ISRE promoter, but not IFN-β-triggered activation of the STAT1 reporter. HEK293 cells were transfected with H1N1-NS2 (200 ng) or empty vector (Vec) and the IFN-β, ISRE or STAT1 luciferase reporter, and indicated plasmids. At twenty hours after transfection, the cells were left untreated or treated with SeV for 12 h or IFN-β for 4 h before reporter assays. (**B**) NS2 proteins from different subtypes of IAVs inhibit SeV-triggered activation of IFN-β promoter. HEK293 cells were transfected with H5N1/H7N9/H9N2-NS2 (200 ng) or Vec and IFN-β luciferase reporter and indicated plasmids. At twenty hours after transfection, cells were left uninfected or infected with SeV for 12 h before reporter assays. (**C**) The effects of overexpressed NS2 on RIG-IN and SeV-induced IFN-β secretion. HEK293 cells were transfected with H1N1-NS2 (200 ng) or Vec, and subsequently transfected with RIG-IN or infected with SeV. The concentration of IFN-β in supernatants was determined using human IFN-β DuoSet ELISA kit. (**D**) NS2 inhibits SeV-triggered transcription of IFN-β and downstream genes. A549 cells were transfected with Flag-NS2 or Vec, and subsequently left uninfected or infected with SeV (MOI = 1) for 12 h before qPCR analysis. (**E**) NS2 facilitates VSV-GFP replication. HEK293 cells were transfected with indicated plasmids for 24 h, and cells were then infected with SeV (MOI = 1) or not (Mock) for 24 h. Cell supernatants were inactivated by ultraviolet radiation for 20 min and collected to treat fresh HEK293 cells for another 24 h. Cells then were infected with VSV-GFP (MOI = 0.01) for 12 h, and then observed microscopically. Scale bar, 1 mm. The data shown represent three independent experiments (*p* < 0.05 (*), *p* < 0.01 (**), *p* < 0.001 (***); ‘ns’ indicates no significant difference).

**Figure 2 viruses-14-02411-f002:**
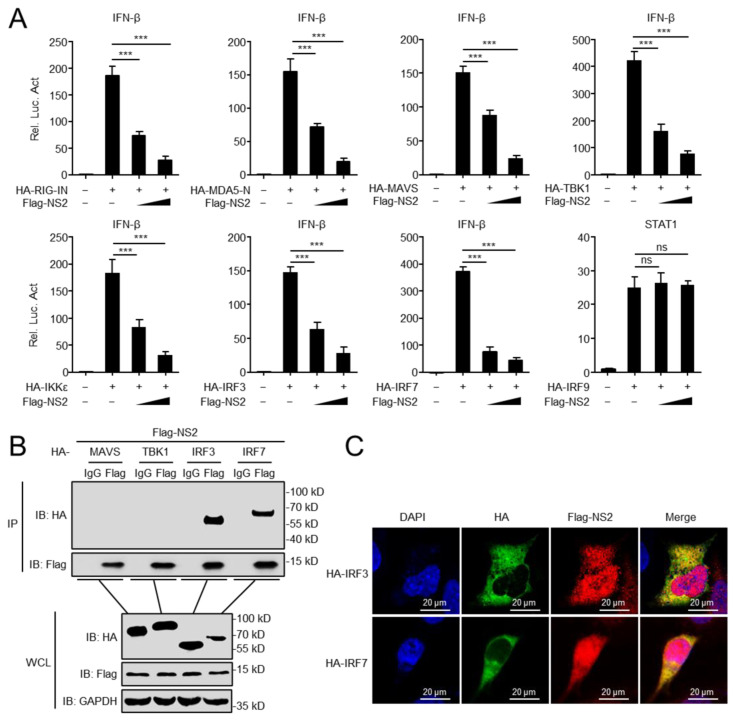
NS2 interacts with IRF3 and IRF7. (**A**) NS2 inhibits activation of the IFN-β promoter induced by RIG-IN, MDA5-Card, MAVS, TBK1, IKKε, IRF3, and IRF7. HEK293 cells were transfected with indicated plasmids along with increasing amounts (0, 200 ng, and 400 ng/mL) of Flag-NS2 expression plasmids. The activation of STAT1-luciferase triggered by IRF9 was used as a negative control. Reporter assays were performed 24 h after transfection. (**B**) Overexpressed NS2 interacts with IRF3 and IRF7. HEK293 cells were transfected with indicated plasmids for 24 h. Then, Co-IP and immunoblotting analyses were performed using indicated antibodies. (**C**) NS2 co-localizes with IRF3 and IRF7 in the cytoplasm. U2OS cells were transfected with Flag-NS2 and HA-IRF3/IRF7 plasmid for 24 h, and then stained with Flag antibody or HA antibody and secondary antibodies. Nuclei were counter-stained with DAPI. Scale bar, 20 μm. The data shown represent three independent experiments (*p* < 0.05 (*), *p* < 0.01 (**), *p* < 0.001 (***), ‘ns’ indicates no significant difference).

**Figure 3 viruses-14-02411-f003:**
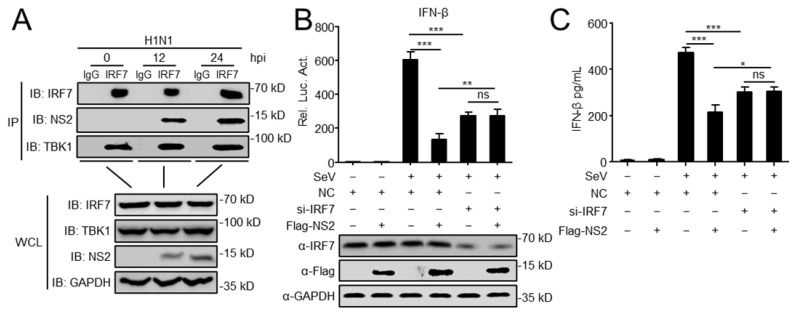
NS2 targets IRF7 and inhibits IFN-I production. (**A**) Endogenous NS2 is associated with IRF7. A549 cells were infected with H1N1 virus (MOI = 0.1) for the indicated times before co-immunoprecipitation and immunoblot analysis. (**B**) Luciferase reporter plasmids (IFNβ-Luc) and the pRL-TK plasmid were co-transfected into HEK293 cells, along with Flag-NS2 and IRF7 siRNA or scrambled siRNA (NC). At twenty hours after transfection, cells were left untreated or treated with SeV for 12 h before reporter assays. (**C**) HEK293 cells were transfected with H1N1-NS2 or Vec, along with Flag-NS2 and IRF7 siRNA or NC for 24 h. Then cells were infected with SeV for 12 h. The concentrations of IFN-β in the supernatants were then detected by using human IFN-β DuoSet ELISA kit. The data shown represent three independent experiments (*p* < 0.05 (*), *p* < 0.01 (**), *p* < 0.001 (***), ‘ns’ indicates no significant difference).

**Figure 4 viruses-14-02411-f004:**
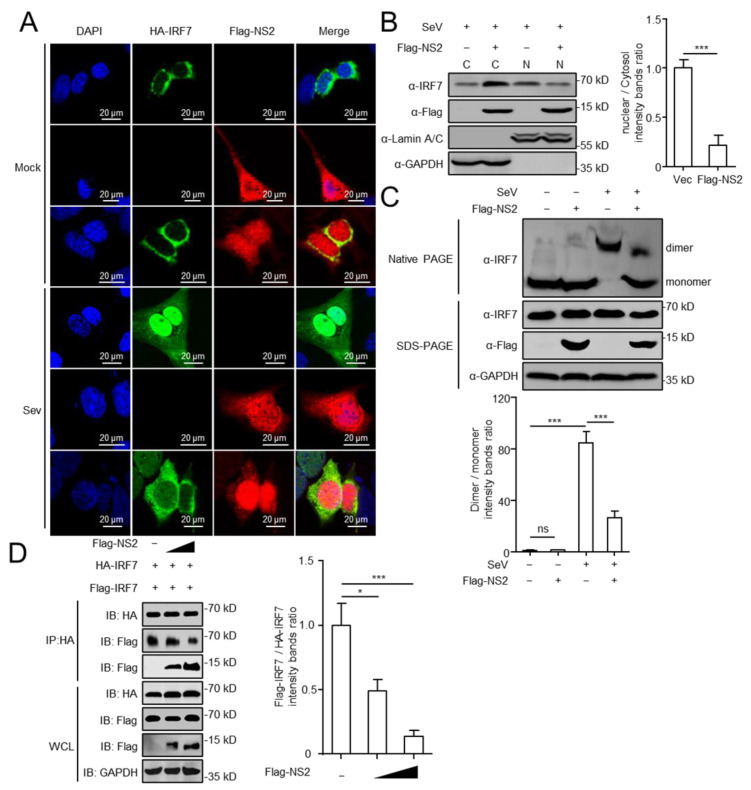
NS2 blocks the nuclear translocation of IRF7. (**A**) U2OS cells were transfected with HA-IRF7 expression plasmid, along with Flag-NS2 or Vec. At twenty hours later, cells were either infected with SeV (4 h) or left untreated. Cells were then stained with rabbit anti-HA and mouse anti-Flag antibodies. Alexa Fluor 488-anti-rabbit immunoglobulin (green) and Alexa Fluor Cy3-anti-mouse immunoglobulin (red) were used as secondary antibodies. Cell nuclei (blue) were stained with DAPI. Scale bar, 20 μm. (**B**) HEK293 cells were transfected with Flag-NS2 or Vec and infected with SeV (4 h) or not. Cell lysates were separated into cytoplasmic and nuclear fractions, and the expression of IRF7 and Flag-NS2 were analyzed by Western blot (fractions: C, purified cytosol; N, purified nucleus). The intensities of the indicated protein bands were determined by using ImageJ (National Institutes of Health, Bethesda, MD, USA), and the ratio of nuclear IRF7 to cytosolic IRF7 was shown in the right panel. (**C**) NS2 suppressed IRF7 dimerization. HEK293 cells were transfected with Flag-NS2 or Vec, and treated with SeV for another 4 h. Native PAGE assays were performed to detect IRF7 dimerization. SDS PAGE assay were performed to detect IRF7, Flag-NS2, and GAPDH. The intensities of the dimer and monomer of IRF7 were determined by using ImageJ (National Institutes of Health, Bethesda, MD, USA), and the ratio of IRF7 dimer to IRF7 monomer was shown in the bottom panel. (**D**) HEK293 cells were transfected with plasmids expressing HA-IRF7 and Flag-IRF7, together with Vec or Flag-NS2 (100 ng and 500 ng) for 24 h. Cell lysates were then immunoprecipitated with anti-HA and immunoblotted with the indicated antibodies. The intensities of the Flag-IRF7 and HA-IRF7 in IP were determined by using ImageJ (National Institutes of Health, Bethesda, MD, USA), and the ratio of Flag-IRF7 to HA-IRF7 was shown in the right panel. The data shown represent three independent experiments (*p* < 0.05 (*), *p* < 0.01 (**), *p* < 0.001 (***)).

**Figure 5 viruses-14-02411-f005:**
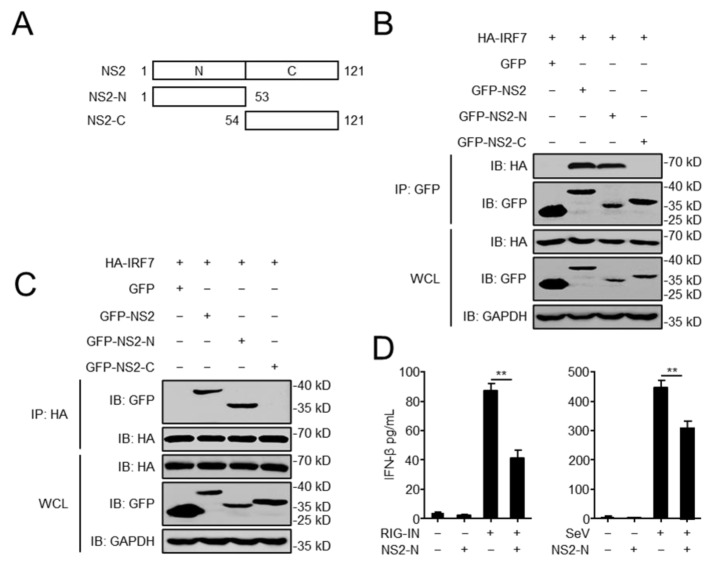
The N-terminal domain of NS2 is essential for the interaction with IRF7. (**A**) GFP-tagged NS2, NS2-N (amino acids (aa) 1 to 53), NS2-C (aa 54 to 121) were constructed and cloned into pEGFP-C1 by using standard molecular biology techniques. (**B**,**C**) The N-terminal domain of NS2 influences the interaction between NS2 and IRF7. HEK293 cells were transfected with HA-IRF7 along with GFP-NS2, GFP-NS2-N, and GFP-NS2-C expression plasmid, followed by IP with anti-GFP or anti-HA. (**D**) The effects of overexpression of the N-terminal domain of NS2 on RIG-IN and SeV-induced IFN-β secretion. HEK293 cells were transfected with GFP-NS2-N or GFP, and subsequently treated with RIG-IN or infected with SeV. The concentration of IFN-β in the supernatants was assessed by using human IFN-β DuoSet ELISA kit. The data shown represent three independent experiments (*p* < 0.05 (*), *p* < 0.01 (**)).

## Data Availability

Not applicable.

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
