# Peer review of "H1N1 Influenza A Virus Protein NS2 Inhibits Innate Immune Response by Targeting IRF7"

_viruses, 2022, doi:10.3390/v14112411_

Round 1
Reviewer 1 Report
In this work, bo zhang et al found that NS2 blocks the nuclear translocation of IRF7 by inhibiting the formation of IRF7 dimers, thereby prevents the activation of IRF7 and inhibits the production of IFN-beta. The work is based on a very strong foundation with clear and robust data, but would benefit with further revision to help strengthen the main conclusions and to better understand.
1. If possible, please provide the Genbank information of the strain in part of method and materials.
2. Figure 1E is confusing, please rewrite related description and figure.
3. No scale bar in Figure 2C.
4. “co-IP” should be “Co-IP”
5. The title is not appropriate, because NS2 also binds with IRF3, which may contribution to the inhibition induced by NS2, not only IRF7.
Reviewer 2 Report
In this manuscript, the authors identified that the nonstructural protein (NS2) of IAV regulates type I IFN signaling. The authors identified that the nonstructural protein (NS2) suppresses the IFNb production by blocking the nuclear translocation of IRF7 by inhibiting the IRF7 dimer formation.
In figure 1, the authors showed that the NS2 is a negative regulator of the type I IFN pathway. The authors tested the effect of over-expressed NS2 on Sendai virus (SeV)-trigged induction of IFNb production and ISG induction. In figure 1A, the authors found that the overexpressed H1N1-NS2 inhibited the Sendai virus (SeV)-triggered activation of the IFNb promoter and ISRE. However, no inhibitory effects were found on STAT1 and STAT3. For type I interferon signaling, STAT1 and STAT2 are crucial mediators. It would be helpful to add the inhibitory effects found on STAT2.
In figure 2, the authors showed data about how NS2 inhibits type I IFN signaling. The authors found that the overexpression of NS2 inhibited the activation of the IFN-β promoter triggered by the expression of RIG-IN, MDA5-N, MAVS, TBK1, IKKε, IRF3, and IRF7 in a dose-dependent manner. However, it had no inhibitory effects on STAT1 activation triggered by IRF9. Again, it would be helpful to add the STAT2 data. Also, in figure 2B, the authors showed the co-IP and immunoblotting analyses. It would be helpful to show the quantitative measurements of the blots.
IN figure 3, the authors showed that the NS2 specifically targets IRF7 and inhibits IFN-I signaling. Furthermore, in figure-4, the authors showed that the N-terminal domain of NS2 is essential for interacting with IRF7. The quantitative measurement of the bands would be helpful.
The virus names were mentioned in lines 31 and 32 as H5Nx and H7Nx IAVs, respectively.
Reviewer 3 Report
In this manuscript, Zhang et al., identified the nonstructural protein 2 (NS2) of IAV as a novel negative regulator of type-I interferon signaling. The authors showed that NS2 could selectively interact with interferon regulatory factor 7 (IRF7) by suppressing the formation of IRF7 dimers, blocking IRF7 nuclear translocation and dampening subsequent interferon production. The conclusion is well-supported by the data presented, and the manuscript is well-written. Nonetheless, some points must be addressed.
Major Points
· Fig 1.A and Line 207, IFN-beta treatment mentioned in the figure legends indicates that cells were stimulated for 12 hours. Why is IFN-beta stimulation required for such a long time?
· In Fig 1.E, the presented data and the text do not explain what is "Vec", and what is the difference between "SeV" and "GFP+SeV", and how MAVS supernatant was produced? Also, this experiment should be well-controlled, quantified, and statistically analyzed.
· For Fig 2.C, it is unclear if the IRF3 and IRF7 observations are presented at the same magnification. In the field presented for IRF7, the nucleus appears to be larger than that presented for IRF3. Also, Fig 4.A needs the same optimization for the magnification as the nucleus size does not appear to be the same.
· Fig 3. B and C need more explanation on how silencing IRF7 could restore the effect of NS2 overexpression. I.e., if NS2 interacts with IRF7 and blocks the nuclear translocation of IRF7, thus suppressing type-I IFN production, IRF7 silencing may further suppress type-I IFN production. How could the authors explain the rescue?
· It would be of importance to identify which of the IRF7 domains can interact with NS2. At least, the authors may discuss this.
·
Minor Points
· Fig 1.B, it should mention how the plasmids of each influenza subtype were constructed.
· Fig 1.C (SEV+NS2) and Fig 3.C (SEV+NS2+si-IRF7), the effect of silencing IRF7 on NS2 effect is ambiguous, with almost the same average between the two panels. Do they have different experimental conditions?
· Fig 1.C and Line 186, the text indicated that NS2 inhibited the IFN-beta production triggered by SeV infection. However, there was no SeV infection in the left panel of Fig 1.C.
· Fig 2.A, the different doses of NS2 were not mentioned, and the protein expression difference was not confirmed by western blotting.
· Fig 2.A, IRF9 treatment was not mentioned in the figure legends. Also, please add literature that supports that IRF9 triggers STAT1 activation.
Reviewer 4 Report
The authors have presented basic mechanism of immune evasion by H1N1 influenza virus. It would be better if they present the findings stating H1N1 and not IAV. As Avian Influenza caused by IAVs may have some differences as compared to H1N1 virus.
Reviewer 5 Report
The authors of this manuscript postulate that the NS2 protein of Influenza blocks the induction of IFN-ß and consequently expression of a luciferase reporter under control of the IFN-ß promoter. The presented data are only partially convincing and lack in parts important controls and soundness. The data on the interaction between IRF7 and NS2 are interesting and convincing, however, how this interaction perturbs the IRF7-mediated induction of IFN-ß is not well supported by the presented data. Several assays lack clarity and suffer from overstatements.
In the light of these short comings, I can not recommend the publication of this manuscript without major revision.
Critical comments:
· The term IFN signaling is not correctly used by the authors. This needs to be improved.
· Lines 176-178: reference DOI: 10.1074/jbc.M114.569178 is missing, please include
Figure 1:
1. The subpanels of the figure are not well separated and overlap, please rearrange to avoid confusion and improve clarity
2. The title of this figure is not correct. It does not show that IFN signaling but IFN induction is blocked by NS2. Reduced IFN-ß and ISRE-reporter activation is likely mediated by reduced IFN-ß induction as shown in subpanel c). This is repeatedly wrongly concluded in the manuscript. Please correct this.
3. Subpanel A) The is no inhibition of the stat1/3-reporter activation by NS2. Please correct in the figure legend (line 204). Also, the STAT-reporter plasmids are not described in the material and methods part. Please describe what promoters are used here?
4. It would be helpful to separate the SeV infection and IFN-ß treated experiments in Fig. 1A into individual subpanels as they reflect different experimental settings.
5. Please include information on the amount of the transfected plasmids in the figure legend as protein functions can be dependent on protein abundance.
6. Line 18018: IFN-ß cannot be “activated”. Please substitute activation with “induction” or “expression”
7. Line 186: IFN-ß production is confusing as it is not specified that this refers to the amount of secreted protein. Please refer to IFN-ß secretion. Overall, the description of the results of subpanel c) is confusing. It is not clear what kind of assay was performed and the assay that is described in the figure legend (ELISA assay was obviously used) is not described in the methods. Please improve and harmonize the description of the performed assays in the legend and the methods part. What is the RIG-IN plasmid for? Please provide some more information on this assay for the uneducated reader.
8. Lines198-199: the results show that NS2 inhibits IFN-ß induction not the IFN-ß signaling pathway. This is two different things. Please correct.
Figure 2:
1. Line 224: interferon induction!
2. Line 225: IFN-ß is not activated but induced!
3. A) What is the trigger to activate signal transduction of the transfected signaling proteins and activate the transcription factors? Expression of the proteins does not readily facilitate their phosphorylation, activation and nuclear translocation. This should require viral RNA or another artificial trigger. If the authors claim that exogenous expression of the proteins activates signaling then they should provide evidence for the activation in the form of western blot or IFA using phospho-site specific antibodies.
4. C) Images of IFR3 and IRF7 single transfection controls is missing. IRF3 seems to majorly interact with NS2 in the nucleus. Why is it then stated that IRF3 and IRF7 interact with NS2 in the cytoplasm? This seems incorrect.
5. Lines 231-233: This sentence is confusing. Interaction of IRF3 and IRF7 was only tested with NS2 not with other proteins. Please rephrase.
6. IRF3 and IRF7 are not part of the IFN signaling pathway but participate in IFN induction
Figure 3:
1. line 251: this is not correct. Liu et al. show that NS2 regulates IFN-ß production, not signaling.
2. Line 254: substitute IFN signaling with IFN-ß expression or induction
3. A) why was TBK1 pulled down and stained? Please describe the purpose of this main text.
4. B and C) This siRNA approach is puzzling and not providing very strong evidence for the interaction of NS2 and IRF7. siRNA KD of IRF7 does not “restore” IFN-ß expression nor promoter activation, since maximum levels are not reaches. The results of this assay should be toned down. “restored” should be substituted with “partially rescued”
Figure 4:
1. A) The NS2 only control is missing in the panel. How long after SeV infection where the cells analysed? After SeV infection, IRF7 translocates to the nucleus. This seems to be inhibited in the presence of NS2. But there is no co-localization of IRF7 and NS2? Rather NS2 translocates completely to the nucleus. Is this also the case without IRF7 overexpression? Please provide the image for this control. How is it envisioned that NS2 blocks IRF7 nuclear translocation without colocalisation?
2. The non-infected control is missing. This is a very important control and would demonstrate to which extent IRf7 and NS2 translocate to the nucleus after infection. In addition this assay needs quantification as the protein levels are only partially shifted and the result are not clear. It would also be helpful to include staining for phosphorylation of IRF7 in this assay as a second measure of activation and functionality.
3. C) please provide quantification of the dimers and provide information on the number of replicates that where performed.
4. D) The conclusion of the authors, that increased levels of co-transfected NS2 prevent IRF7 dimerization is intriguing but not precisely supported by the shown blot. The Flag bands appears to suffer from an air bubble or improper transfer as they are not well displayed. The assay and blot should be repeated and it would help to include more concentrations of NS2 to see the claimed dose-dependency. Also, IRF7 phosphorylation ad dimerization takes place in the cytoplasm. How can NS2 prevent IRF7 dimerization when it is in the nucleus (see fig. 4A)?
Round 2
Reviewer 2 Report
The authors revised the manuscript to respond to the comments.